# Moral Distress in Oncology: A Descriptive Study of Healthcare Professionals

**DOI:** 10.3390/ijerph20085560

**Published:** 2023-04-18

**Authors:** Lara Guariglia, Irene Terrenato, Laura Iacorossi, Giovanna D’Antonio, Sonia Ieraci, Stefania Torelli, Fabiola Nazzicone, Fabrizio Petrone, Anita Caruso

**Affiliations:** 1Psychology Unit, IRCCS Regina Elena National Cancer Institute, Via Elio Chianesi, 53, 00144 Rome, Italy; 2Biostatistics & Bioinformatics, IRCCS Regina Elena National Cancer Institute, 00144 Rome, Italy; 3Nursing Research Unit IFO, IRCCS Regina Elena, National Cancer Institute, 00144 Rome, Italy; 4Nursing, Technical, Rehabilitation, Assistance, and Research Direction IFO, IRCCS Regina Elena, National Cancer Institute, 00144 Rome, Italy

**Keywords:** moral distress, organizations, moral distress scale, nurses, physicians, ethics

## Abstract

Background: The oncology setting is characterized by various complexities, and healthcare professionals may experience stressful conditions associated with ethical decisions during daily clinical practice. Moral distress (MD) is a condition of distress that is generated when an individual would like to take action in line with their ethical beliefs but in conflict with the healthcare facility’s customs and/or organization. This study aims to describe the MD of oncology health professionals in different care settings. Methods: Descriptive quantitative study was conducted in the Operating Units of the Istituti Fisioterapici Ospitalieri in Rome between January and March 2022. The investigated sample consisted of the medical and nursing staff on duty at the facility, who were given a questionnaire through a web survey. Besides a brief sociodemographic form, the MD Scale-Revised questionnaire was used for data collection. Results: The sample consisted of nurses (51%) and physicians (49%), predominantly working in surgeries (48%), and having 20–30 years of service (30%). MD was higher among healthcare professionals, in medicine than that ing in corporate organizations, surgeries, or outpatient clinics (*p* = 0.007). It was not related to the profession (*p* = 0.163), gender (*p* = 0.103), or years of service (*p* = 0.610). Conclusions: This paper outlines the prevalence of MD in care settings and describes its relationship with profession, gender, and seniority. There is no patient care without the care of health professionals: knowing and fighting MD improves the safety of the treatments provided and the quality perceived by patients.

## 1. Introduction

Oncology is considered one of the largest areas and encompasses many competencies that healthcare professionals can and should perform in an increasingly complex and ever-changing environment [1]. Moreover, the organization of the healthcare system presupposes that its operators make decisions regarding the definition of priorities in their daily work, make ethically based choices on therapeutic and welfare issues and the different present and future needs of the patient, and take into account the objectives and standards of their structure. Furthermore, current trends are moving toward reduced staffing, increased patient care complexity, and the need to maintain appropriate clinical practice standards in a context increasingly focused on reducing healthcare costs. In this scenario, it is not surprising to experience stressful situations associated with ethical decisions during daily clinical practice [2]. Indeed, several studies have shown how changes in healthcare management have added new stressors to the healthcare profession [3]. Several years ago, the scientific literature identified a specific stressor related to healthcare providers’ distress when they could not adequately meet their perceived responsibilities: Moral distress (MD [4]).

MD was first defined by Jameton (1984) [5], who described it as “a psychological imbalance that occurs when healthcare professionals are aware of the most appropriate moral action for the situation but cannot perform it because of institutional obstacles or power hierarchies”. Over time, this description has been subject to different redefinitions and articulations. Today, moral distress is defined, in a more synthetic way, as the discomfort that healthcare professionals feel when they are unable to face their perceived responsibilities and must, therefore, make choices with implications on an ethical and personal level [5,6,7].

The various studies conducted over time have shown that other factors, in addition to organizational factors, are sources of MD: powerlessness or lack of adequate knowledge, clinical situations such as aggressive treatments given to terminally ill patients, pain management patterns, maintenance of the appropriate patient quality of life standards, and conflicts between staff, patient and family members [6,7,8,9,10].

MD can be expressed as psychological distress, fatigue, disengagement, or alienation from the profession [11].

It is a complex phenomenon frequently experienced in departments where ethical conflicts are linked to technological progress and high-intensity work environments, or when operators are involved in decisions concerning the end of life. It can, on the one hand, be defined as healthcare personnel being required to have ever greater competence about the ability to make ethically valid decisions aimed at greater patient protection, and on the other hand, as these staff themselves maybe lack both resources and individuals, as well as lacking adequate coordination within the staff and organizational resources. 

MD has implications not only for the well-being of the individual, but also for care outcomes and, more generally, for the entire social and healthcare system; it is not uncommon for healthcare professionals experiencing a situation of stress and emotional exhaustion to express their anger at a colleague or patient, and even to adopt an attitude of “avoidance” toward the conflict situation, denying all care provision and nurturing the idea of leaving the profession [12]. In some cases, albeit less frequently, the healthcare provider tends to counteract the negative experience by overcompensating with excessive care delivery [12].

Accordingly, the literature reports how human resources play a strategic role because they can influence the outcome of the care process; therefore, an institute needs to identify the phenomenon of MD among healthcare providers at an early stage through specific rating scales. Among all healthcare professionals, the nurse appears to be the most at risk of developing MD, mainly when employed in critical areas [13]. Furthermore, a cross-sectional study conducted in Brazil investigated the correlation between job satisfaction and MD, showing an inversely proportional relationship [14]. The same study also found that distress levels were higher in newly hired nurses and those working in the intensive care unit [14]. Oncology is also one of the areas most at risk for the development of MD, which is saturated with ethical dilemmas arising from the terminally ill condition of patients and the administration of seemingly “unnecessary” therapies [13,15]. A study by G. Jones [16] conducted on nurses in the oncology area highlighted among the most frequent causes of MD a multiplicity of ethical dilemmas, among which emerged, in more detail, the pain management method, problems related to cost containment and the maintenance of adequate standards of quality of life for the patient. The departments with a high incidence of this problem also appear to be nephrology and dialysis.

To date, most of the quantitative studies in the literature have been conducted in intensive care units and on nursing staff. Studies investigating the prevalence of MD in medical staff involved in the care of oncology patients are still few. Therefore, this study aims to describe the MD of healthcare professionals (physicians and nurses) working in oncology in different clinical settings. Evaluating this aspect is essential for understanding the problem and taking proper corrective measures based on a more elastic organizational management view.

## 2. Materials and Methods

This quantitative descriptive study was conducted in the Operating Units of the Istituti Fisioterapici Ospitalieri (IFO) in Rome between 18 January and 24 March 2022. The investigated sample consisted of the medical and nursing staff on duty at the facility, who were given a questionnaire through a web survey. Inclusion criteria for recruitment were as follows: being healthcare professionals (physicians and nurses) employed at the IFO and adhering to the present study by completing informed consent forms. Within the IFO, other medical and nursing healthcare operators deal with non-cancer patients, are there in training but only for a short time, or belong to outsourcing companies that do not have an institutional email, a factor which would not permit the photographing of the figures involved in their daily professional activity for the focus of our attention. Therefore, the criteria for excluding the sample were precise: physicians and nurses employed in the care of non-oncology patients; physicians and nurses in specialty training at the IFO; healthcare professionals from outsourcing companies or freelancers who did not have an institutional email address, such as students. Before the start of the study, the IFO Central Committee’s approval was obtained (EC Register of Trials RS1591/21 dated 21 September 2021), and the study’s purpose and participation were explained to participants via email.

For organizational reasons, excluding a priori healthcare professionals not directly involved in oncology patient care was not possible, although they did not meet the study criteria. This part of the population was excluded retrospectively when responses were received.

In addition to the administration of the questionnaire, the following data were collected: unit of service, gender, and years of service: (1–10 years; 10–20 years; 20–30 years; >30 years).

The MD Scale-Revised questionnaire (MDS-R) [17], administered as a web survey and sent to participants via an institutional email address, was used for data collection in addition to a short socio-demographic form. Data collection was done by preserving confidentiality (privacy and anonymity).

The MDS-R is a questionnaire with 14 items investigating four components: the futility of care, perception of unethical behavior, communication between staff and families, and poor teamwork [17]. Using a Likert scale from 0 to 4, respondents were asked to rate each of the 14 items on the frequency of the event experienced and how uncomfortable it was in terms of intensity. The total MDS-R score ranges from 0 to 16 and is obtained by summing the frequency multiplied by intensity scores and dividing the total by the number of items. In addition to the 14 items, the scale consists of two more questions addressing attitudes toward the job. The two questions investigate whether the healthcare provider has thought about leaving their job in the past or present.

## 3. Statistical Analysis

Descriptive statistics were produced for all variables of interest. Categorical variables were summarized using frequencies and percentage values, and continuous variables by median values and their range. Nonparametric Kruskal–Wallis or Mann–Whitney tests were applied to analyze potential differences between groups when appropriate. A *p*-value < 0.05 was considered statistically significant. All statistical analyses were performed with SPSS software (SPSS version 21, SPSS Inc., Chicago, IL, USA).

## 4. Results

The questionnaire was sent to a total of 677 healthcare professionals. Overall, 299 healthcare professionals (44%) joined the study and completed the questionnaire upon acceptance of informed consent. Out of the total number of questionnaires collected, 29 (9.7% of the total) were not analyzed for the study because they were filled out by healthcare providers not directly involved in the care of oncology patients. The total number of questionnaires surveyed was 270. The sample consisted of nurses (51%) and physicians (49%), predominantly female (66%), working in surgeries (48%), and with 20–30 years of service (30%) (Table 1).

### 4.1. Moral Distress and Clinical Settings

We measured MD in the different facilities by grouping the operating units into five main categories: outpatient clinics, medicine, surgeries, diagnostics, and corporate organization. The results showed a significant difference in the overall score at the MSD-R (*p* = 0.001) among the different affiliation facilities. In the specific comparison between individual facilities, a significant difference in the global score at MSD-R emerged between medicine and corporate organizations (*p* = 0.034) for which the level of MD was higher among healthcare providers working in medicine than in corporate organizations; between medicine and outpatient clinics (*p* = 0.007), the level of MD was higher among healthcare professionals working in medicine than in outpatient clinics; between medicine and surgeries (*p* = 0.010), the level of MD was higher among healthcare professionals working in medicine than in surgeries. Table 2 shows the median values of the overall score at the MSD-R in the different facilities.

Based on the rating scale items analysis, a significant difference was found between the affiliated facilities in the Futility Rate and Deceptive Communication subscales. Regarding the Futility Rate, it was found that the five units had a different distribution of scores; however, while we obtained a significant difference overall (*p* = 0.037), no significant difference emerged in the pairwise subscales. Regarding Deceptive Communication, an overall significant difference resulted between the different facilities (*p* = 0.008) and in the sub-comparison between surgeries and outpatient clinics (*p* = 0.024) and between medicine and outpatient clinics (*p* = 0.006). In Deceptive Communication, surgeries and medicine had higher MD levels than in outpatient clinics. Table 2 shows the median Futility Rate and Deceptive Communication score in the different facilities. Poor Teamwork showed no significant differences in the overall score at MSD-R (*p* = 0.169).

### 4.2. Moral Distress, Profession, Gender, and Years of Service

We compared the MD scores of male and female physicians and nurses and their years of seniority. The results showed no significant differences between physicians and nurses in the overall MDS-R score (*p* = 0.163) or the individual scales: Futility Rate (*p* = 0.105), Ethical Misconduct (*p* = 0.122), Deceptive Communication (*p* = 0.168), Poor Teamwork (*p* = 0.061). No significant differences arose either between males and females in the overall MDS-R score (*p* = 0.103) or the individual scales: Futility Rate (*p* = 0.821), Ethical Misconduct (*p* = 0.221), Deceptive Communication (*p* = 0.076), and Poor Teamwork (*p* = 0.245).

Similarly, the comparison of years of seniority in the overall MDS-R score (*p* = 0.610) and the individual scales showed no statistically significant differences: Futility Rate (*p* = 0.513), Ethical Misconduct (*p* = 0.430), Deceptive Communication (*p* = 0.828), and Poor Teamwork (*p* = 0.585) (Table 3).

### 4.3. Moral Distress and Job Position

In analyzing the relationship between MD and job position, to the question, “Have you ever thought about leaving your job because of the ethical distress experienced in your work environment?”, it was found that those who had thought about leaving or had left their jobs in the past had a significantly higher score on the overall scale (*p* = 0.018) than those who had never thought about leaving their employment. In the subscale analysis, significant differences resulted between these two groups regarding Ethical Misconduct (*p* = 0.008) and Poor Teamwork (*p* < 0.001). In the difference between genders, males reflected the same trend as the total sample (overall scale, *p* = 0.015; Ethical Misconduct, *p* = 0.002; Poor Teamwork, *p* < 0.001); among females, there was no difference in the score on the overall scale, but a significant difference was found in Poor Teamwork (*p* < 0.001). When we compared physicians and nurses, it emerged that physicians who in the past had thought of leaving or had left their job had higher scores on the overall scale (*p* = 0.049) in the Ethical Misconduct (*p* = 0.018) and Poor Teamwork (*p* = 0.019) subscales. A significant difference was found only among nurses in the Poor Teamwork subscale (*p* = 0.001). When we related job position to years of seniority, it emerged that those with 0–10 years of service had a higher score in the Poor Teamwork subscale (*p* = 0.031), those who have worked 10–20 years had a higher score in the Deceptive Communication subscale (*p* = 0.030), and those who have worked 20–30 years had higher scores in Ethical Misconduct (*p* = 0.005), in Deceptive Communication (*p* = 0.047), and Poor Teamwork (*p* = 0.013). Finally, those working for more than 30 years had higher values in Poor Teamwork (*p* = 0.006).

In the relation between job positions and different facilities, significant differences resulted between those who thought about leaving or quitting their job and those who never thought about leaving diagnostics or medicine. In diagnostics, differences were found in the overall score (*p* = 0.007), Ethical Misconduct (*p* = 0.001), Deceptive Communication (*p* = 0.018), and Poor Teamwork (*p* = 0.001). In medicine, differences appeared in Ethical Misconduct (*p* = 0.021) and Poor Teamwork (*p* = 0.002). Regarding the question “Are you currently thinking of leaving your job?”, the only notable finding resulted in diagnostics, where those thinking of leaving their employment had higher scores in Ethical Misconduct (*p* = 0.043) (Data upon request.)

## 5. Discussion

The study aimed to describe the MD of medical and nursing staff in oncology in different settings. The studies carried out to date to measure moral distress in oncology have been conducted on specific categories or professional fields, such as nurses or pediatric oncology, or are qualitative. Our study gives us a relatively comprehensive picture of the perception of MD in different healthcare categories of affiliation, gender, and years of practice.

It appears that the MD level does not change depending on the type of profession practiced, gender, or years of seniority. Concerning the findings on the lack of differences between physicians and nurses, this result confirms what emerged in the validation work of the MDS-R scale in Italy [17]. However, it differs from other studies in [11,13], where nurses expressed a higher level of MD than physicians. The absence of significant differences in the perception of moral distress between doctors and nurses, in the context of cancer treatment, could be explained in two different ways, referring to the literature. The study by Lamiani et al. (2022) [17], demonstrates how, in the context of pediatric care, factors of an organizational nature, such as the number of hospitalizations, the relationship with patients, and staff shortages, play a more relevant role than factors of an individual nature in the perception of a higher level of moral distress. In this sense, it is conceivable that these factors influence all healthcare personnel equally. In other studies, however, apparently, conflicting results emerge. In the study by Whitehead et al. (2015) [13], nurses expressed a higher level of moral distress as they were involved in direct patient care. In the study by Rodriguez-Ruiz et al. (2022) [18], it is doctors experienced a higher level of moral distress over the lack of continuity and efficiency of care. Based on these two studies, it can be hypothesized that there are factors that weigh specifically on the different professional categories (direct care for nurses, responsibilities for doctors), but which make both groups sensitive to the perception of moral distress.

Studies investigating the relationship between MD and years of practice reported mixed results. Some studies recorded a higher level of MD in healthcare providers with more years of practice [6,19,20], as repeated exposure to MD experiences would increase the perception of distress. Other studies found that nurses lacking experience in dealing with ethically challenging situations at the beginning of their professional careers may be at greater risk of MD [20]. In contrast, no significant differences were found in further studies [21]. Our study found no significant differences between the relation of MD with years of practice among physicians or nurses.

Few studies analyzed MD regarding the various structures that healthcare providers are affiliated with. Sirilla’s study [22] showed that MD is present in nurses working in oncology regardless of the specific unit. Our study revealed interesting differences between the different facilities to which healthcare providers are affiliated. Healthcare professionals working in medicine had higher levels of distress than those working in surgeries, outpatient clinics, and corporate organizations. This finding can be explained by the fact that, in medicine, hospital stays are likely to be longer during treatment and while critical events or advanced stages of illness are being managed; therefore, healthcare providers are more frequently confronted with issues such as communication with the patient and their families, aggressive care, and pain management, thus developing greater sensitivity [23]. MD differences were mainly observed between facilities in Futility Rate and Deceptive Communication: for the former, an overall difference emerged between units; for the latter, medicine and surgeries had higher MD levels than outpatient clinics. Futility Rate refers to interventions that rarely benefit the patient and, in most cases, delay end-of-life decisions, such as referral to palliative care services [24].

Thus, it is obvious how the application of futile care to the patient is directly related to the difficulty in establishing a clear communication plan with the patient at different stages of the disease. In an oncology facility, these are sensitive and complex issues that burden the staff’s sense of responsibility and ethics; lack of training, family caregiving expectations, and power hierarchies that limit individual capacity and burden in terms of emotional distress are not uncommon.

The last result of our work concerns the relationship between the intention to quit and high MD scores. According to the study by Austin et al. (2017) [2], it appears that the conditions contributing most to the perception of high-level MD and job abandonment by physicians and nurses are the provision of futile and inappropriate care and a high number of patients in intensive care units. A significant correlation between the intention to resign from employment and MD was also found in the study by Hashemi et al. (2020) [24]. Our study confirms previous works and shows that healthcare professionals who had considered leaving their job had significantly higher levels of MD than others. One of the most interesting data emerged from the subscale survey; Poor Teamwork was found to be the aspect in which the MD level was significantly higher among those who had decided to leave or had left their job in the following categories: males, females, physicians, nurses, those with 0 to 10 years, 20 to 30 years, and more than 30 years of service, those working in medicine and diagnostics. Therefore, it seems clear that Poor Teamwork is among the MD factors with the greatest impact on job retention. In fact, within the work team, situations causing MD can easily go unrecognized, be minimized, or be inefficiently addressed [25]. Additionally, many elements can foster the perception of MD within the work team: decision-making hierarchy, the attitude of obedience, the conflict between authority and perceived professional obligations rather than lack of real authority, poor ability to cooperate, or poor communication skills [25,26].

The second relevant item is Ethical Misconduct, which achieved significantly higher scores in the overall difference between those who had decided to leave or had left their jobs and those who had not thought of doing so or done so, specifically for the following categories: physicians, males, those working in their 20s to 30s, and those working in medicine. This is a factor that has already been explored in the literature in terms of its various constituent components: the burden of inadequate care provided by other healthcare professionals, the weight of the decision-making hierarchy [27], and the difficulty of managing the relationship with family members [28].

The study’s limitations concern being single-center and the lack of comparison between the sample of 44% who answered the questionnaire and the remaining 66%who did not join the study. Given that the study was conducted on the institution’s in-house staff, we maintained a tremendous focus on privacy, and their demographic age is not even available; therefore, we do not have data that would allow us to make a comparison between the two groups.

## 6. Conclusions

The purpose of this study was to describe the MD of healthcare professionals (physicians and nurses) working in oncology in different clinical settings. The results showed that levels of MD increased in healthcare professionals working in medical units. MD does not change based on the type of profession practiced, gender, or years of seniority, while Poor Teamwork leads to job abandonment. The data that emerges on Poor Teamwork deserves attention because, in some ways, it represents one of the key elements of the problem: when a professional take an ethically relevant decision, even if judged formally consistent with presumed professional duties, this does not mean that it does not involve any cost in terms of emotions and feelings about oneself or the other subjects involved. In this sense, the decision-making process should be conceived and carried out within an open communication and coordination system. This figure deserves more in-depth investigations of the medical and nursing staff and adequate support interventions.

Despite the need for further investigation, the data acquired from the investigated Italian population may help us to better understand the problem and help the oncology team.

## Figures and Tables

**Table 1 ijerph-20-05560-t001:** Characteristics of the sample.

	N.	%
Profession		
Physician	133	49.00
Nurse	137	51.00
Gender		
Male	91	34.00
Female	179	66.00
Unit		
Outpatient clinics	22	8.00
Surgeries	131	48.00
Diagnostics	51	19.00
Medicine	53	20.00
Corporate Organization	13	5.00
Years of work		
0–10	70	26.00
10–20	50	19.00
20–30	82	30.00
30+	68	25.00

**Table 2 ijerph-20-05560-t002:** Summary of median values with the stratified analysis of statistically significant comparisons.

Item	Median (Min–Max)	*p*-Value
Overall score	3.50 (0–36.50)	0.001 * (overall)
Outpatient clinics	2.29 (0–22.75)	outpatient clinics vs. medicine 0.007
Surgeries	3.35 (0–36.5)	surgeries vs. medicine 0.010
Diagnostics	4.25 (0–32.00)	
Medicine	5.00 (0.92–30.00)	
Corporate organization	2.71 (0.64–8.50)	corporate org. vs. medicine 0.034
Futility Rate	2.63 (0–14.66)	0.037 * (overall)
Outpatient clinics	0.83 (0–12.00)	
Surgeries	2.66 (0–13.33)	
Diagnostics	1.33 (0–10.66)	
Medicine	3.33 (0–14.66)	
Corporate organization	1.66 (0–9.00)	
Ethical Misconduct	1.60 (0–9.60)	0.143 * (overall)
Outpatient clinics	1.20 (0–5.50)	
Surgeries	1.60 (0–9.60)	
Diagnostics	1.20 (0–8.80)	
Medicine	2.00 (0–9.00)	
Corporate organization	0.80 (0–9.40)	
Deceptive Communication	2.66 (0–12.00)	0.008 * (overall)
Outpatient clinics	1.33 (0–6.33)	
Surgeries	3.00 (0–11.30)	outpatient clinics vs. surgeries 0.024
Diagnostics	2.33 (0–10.66)	
Medicine	3.00 (0–12.00)	outpatient clinics vs. medicine 0.006
Corporate organization	3.00 (0.33–11.00)	
Poor_Teamwork	4.00 (0–16.00)	0.169 * (overall)
Outpatient clinics	4.00 (0–12.00)	
Surgeries	4.30 (0–16.00)	
Diagnostics	2.0 (0–16.00)	
Medicine	4.66 (0.66–12.66)	
Corporate organization	4.00 (0–10.66)	

* Kruskal–Wallis test.

**Table 3 ijerph-20-05560-t003:** Stratified analysis (by profession, gender, and years of service) for each item.

Item	Profession	Gender	Years of Service
	Physician vs. Nurse	Female vs. Male	0–10 Years vs. 10–20 Years vs. 20–30 Years vs. 30+ Years
	*p*-Value Test Mann–Whitney	*p*-Value Test Mann–Whitney	*p*-Value Test Kruskal–Wallis
Overall Score	0.163	0.103	0.610
Futility Rate	0.105	0.821	0.513
Ethical Misconduct	0.122	0.221	0.430
Deceptive Communication	0.168	0.076	0.828
Poor_Teamwork	0.061	0.245	0.585

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
