# Peer review of "Moral Distress in Oncology: A Descriptive Study of Healthcare Professionals"

_ijerph, 2023, doi:10.3390/ijerph20085560_

Round 1

Reviewer 1 Report

Dear authors

First of all, my congratulations on this work. 

After my review, my suggestions and comments are:

1) Methods

The exclusion criteria need to be explained. If there is "moral distress" when there is a conflict between what one intends to do and the organisation's limitations, this situation may also affect other professionals. They could even be used to analyse differences.

2) Results

a) The sampling error must be presented;

b) Replace "more distress" with "moral distress" in the first section;

c) The descriptive statistics of the score values of the scale and its domains must be presented before the analysis.  To analyse without presenting the global scores does not show us the overall scores (general and domains); this comment refers to all sections of the results;

3) Discussion 

You refer that your study did not show differences between nurses and physicians as in other studies. So you must try to explain why you did not find it.

Author Response

Authors’ response to Reviewer’s comments

We are deeply grateful to the Reviewers and the Editor for the time devoted to our work and for their valuable suggestions that we believe helped to improve it. Our detailed responses follow each comment.

Reviewer 1

1)Methods

Reviewer Comment: “The exclusion criteria need to be explained. If there is "moral distress" when there is a conflict between what one intends to do and the organisation's limitations, this situation may also affect other professionals. They could even be used to analyse differences”

Author Response to Comment: We agree with the reviewer and we added in Materials and Methods: “Within the IFO there are other medical and nursing healthcare operators who deal with non-cancer patients, or who are in training but only for a short time, or who belong to outsourcing companies that do not have institutional email, which would not allow to photograph the figures really involved in their daily professional activity to our focus of attention. Therefore, the criteria for excluding the sample were precisely: doctors and nurses employed in the care of non-oncological patients; doctors and nurses in specialist training at IFO; healthcare professionals from outsourcing companies or freelancers who did not have an institutional email address, such as students”.

2) Results

Reviewer Comment: a) “The sampling error must be presented”

Author Response to Comment: For all variables of interest we reported the median values with their relative ranges, since we tested the distributions with non-parametric tests. Calculate the standard error in not appropriate in this context.

Reviewer Comment: b) Replace "more distress" with "moral distress" in the first section;

Author Response to Comment: done

Reviewer Comment: c) The descriptive statistics of the score values of the scale and its domains must be presented before the analysis. To analyse without presenting the global scores does not show us the overall scores (general and domains); this comment refers to all sections of the results.

Author Response to Comment: We added the overall scores for all domain in tab.2

3) Discussion

 Reviewer Comment: “You refer that your study did not show differences between nurses and physicians as in other studies. So you must try to explain why you did not find it”.

Author Response to Comment: We agree with the reviewer and we added in the discussion: “The absence of significant differences in the perception of moral distress between doctors and nurses, in the context of cancer treatment, could be explained in two different ways, referring to the literature. The study by Lamiani et al. (2022) demonstrates how, in the context of pediatric care, factors of an organizational nature, such as the number of hospitalizations, the relationship with patients, and staff shortages, play a more relevant role than factors of an individual nature in the perception of a higher level of moral distress. In this sense, it is conceivable that these factors influence all healthcare personnel equally. In other studies, however, apparently conflicting results emerge. In the study by Whitehead et al. (2015), nurses expressed a higher level of moral distress as they were involved in direct patient care. In the study by Rodriguez-Ruiz et al. (2022), it is the doctors who perceive a higher level of moral distress about the lack of continuity and efficiency of care. Starting from these two studies, it can be hypothesized that there are factors that weigh specifically on the different professional categories (direct care for nurses, responsibilities for doctors) but which make both groups sensitive to the perception of Moral Distress”

Reviewer 2 Report

First congratulate the research team, the research is relevant and it is important to identify and prevent DM since situations that affect mental health can have serious complications in the future.

Author Response

Authors’ response to Reviewer’s comments

We are deeply grateful to the Reviewers and the Editor for the time devoted to our work and for their valuable suggestions that we believe helped to improve it. Our detailed responses follow each comment.

Reviewer 2

Reviewer Comment: First congratulate the research team, the research is relevant and it is important to identify and prevent DM since situations that affect mental health can have serious complications in the future.

Author Response to Comment: We thank the reviewer. We agree with the reviewer. We too believe it is very important to identify and prevent MD as situations that affect mental health can have serious complications in the future.